# Cardiovascular risk in subjects over 55 years of age and cognitive performance after five years. NEDICES2-RISK study. Study protocol

Ester Tapias-Merino[1,2,3,4]*, María del Canto De Hoyos-Alonso[5], Israel Contador-Castillo[6], Emiliano Rodríguez-Sánchez[2,7,8], Teresa Sanz-Cuesta[9,10], Concepción María Becerro-Muñoz[2,7], Jesús Hernández-Gallego[4,11,12], Saturio Vega-Quiroga[13], Félix Bermejo-Pareja[11,12,14], NEDICES2-RISK Group[¶]

1 Healthcare Centre Comillas, Gerencia Asistencial Atención Primaria, Servicio Madrileño de Salud, Madrid, Spain, 2 Spanish Research Network for Preventive Activities and Health Promotion in Primary Care (REDIAPP), Spain, 3 Research Institute Hospital 12 de Octubre i+12, Group for Research in Health Services and Results, Madrid, Spain, 4 Faculty of Medicine, Department of Medicine, Complutense University of Madrid, Madrid, Spain, 5 Healthcare Centre Pedro Laín Entralgo, Gerencia Asistencial Atención Primaria, Servicio Madrileño de Salud, Madrid, Spain, 6 Faculty of Psychology, Department of Basic Psychology, Psychobiology and Methodology of Behavioral Sciences, University of Salamanca, Salamanca, Spain, 7 Research Unit for Primary Health Care, Institute for Biomedical Research of Salamanca (IBSAL), Castilla y León Health Service, Salamanca, Spain, 8 Department of Medicine, University of Salamanca, Salamanca, Spain, 9 Research Unit, Gerencia Asistencial de Atención Primaria (GAAP), Madrid, Spain, 10 Health Services Research on Chronic Patients Network (REDISSEC), Instituto Salud Carlos III, Madrid, Spain, 11 Department of Neurology, Hospital Universitario 12 de Octubre (Madrid, Spain), 12 Research Institute Hospital 12 de Octubre i+12, Neurosciences Group CIBERNED, Madrid, Spain, 13 Healthcare Centre Arévalo, Gerencia Asistencial Atención Primaria, Sanidad de Castilla y León, Arévalo, Spain, 14 Consulting Neurologist, Hospital 12 de Octubre, Madrid, Spain

¶ Membership of NEDICES2-Risk Group is provided in the Acknowledgments
* ester.tapias@salud.madrid.org

Data Availability Statement: Regarding data exchange, the Ethics Committee for Research of

## Abstract

### Background

Cognitive impairment and dementia have a high prevalence among the elderly and cause significant socio-economic impact. Any progress in their prevention can benefit millions of people. Current data indicate that cardiovascular risk (CVR) factors increase the risk of developing cognitive impairment and dementia. Using models to calculate CVR specific for the Spanish population can be useful for estimating the risk of cognitive deterioration since research on this topic is limited and predicting this risk is mainly based on outcomes in the Anglo-Saxon population. The aim of this study is to assess the relationship between CVR in the Spanish population, as calculated using the FRESCO (Función de Riesgo Española de acontecimientos Coronarios y Otros) and REGICOR (Registre Gironí del Cor) CVR tables, and the change in cognitive performance at a 5-year follow-up.

### Methods

**Design**: Observational, analytic, prospective cohort study, with a 5-year follow-up. **Ambit**: Population. **Population**: Subjects 55 to 74 years of age, included in the NEDICES2 (2014–2017) cohort, who did not present dementia and had undergone the neuropsychological

the Hospital Universitario 12 de Octubre approved this research without considering the option of data sharing. The data contains sensitive clinical information about the patient, so there are ethical and legal restrictions to sharing the data set. The data are part of the NEDICES2-RISK study and can be requested by contacting the Primary Health Care Research and Innovation Foundation (FIIBAP) in the Community of Madrid at the email address fiibap@salud.madrid.org for the request of data.

**Funding:** ETM. The funding body of this study is Instituto de Salud Carlos III (project reference PI18/00522), with co-funding from the European Regional Development Fund "A way to make Europe". The project has been peer-reviewed by the funding agency. https://eng.isciii.es/eng.isciii.es/Paginas/Inicio.html. ETM. ". It received a grant for research activity from the Fundación para la Investigación e Innovación Biomédica de Atención Primaria (FIIBAP) in the Community of Madrid via their call for grants in 2018 and 2019. Funders had and will not have a role in study design, data collection and analysis, decision to publish, or preparation of the manuscript. https://www.fiibap.org/.

**Competing interests:** The authors have declared that no competing interests exist.

**Abbreviations:** CES-D, Center for Epidemiological Studies Depression Scale; EuroQol-5D, European Quality of Life scale; FAQ, Pfeffer's functional activities questionnaire; FRESCO, Función de Riesgo Española de acontecimientos Coronarios y Otros; MEDLIFE index, Mediterranean lifestyle index; MMSE-37, 37-item version of the Mini-Mental State Examination; GPPAQ, General practice physical activity questionnaire; NEDICES, NEurological DIsorders in CEntral Spain; CR, cognitive reserve; CVR, cardiovascular risk; REGICOR, Registre Gironí del Cor; TMT, Trail Making Test.

evaluation (N = 962). **Variables**: Exposure factors (CVR factors and estimated risk according to the CVR predictors by REGICOR and FRESCO), dependent variables (change in the score of the brief neuropsychological test in the study NEDICES2 five years after the first evaluation), and clinical and socio-demographic variables. **Statistical analysis**: Analysis of data quality. Descriptive analysis: socio-demographic and clinical variables of subjects. Bivariate analysis: relationship between basal CVR and change in neuropsychological tests. Multivariate analysis: relationship between basal CVR and change in neuropsychological tests adjusted by co-variables. Analysis and comparison of the reliable change in independent samples.

## Discussion

The Spanish population can benefit from determining if individuals with high CVR, which is commonly detected in usual clinical practice, will present decreased cognitive performance compared to subjects with lower CVR. This study can affect how to address CVR factors and the design of effective prevention strategies for cognitive deterioration.

## Trial registration

Clinicaltrials.gov, NCT03925844.

## Background

Population aging leads to an increase in the prevalence of chronic conditions, including neurodegenerative diseases and dementia in particular. Chronic illnesses are emerging as the main cause of death in developed countries and are at the origin of the majority of disabilities that individuals suffer and neurological disorders contribute to a significant proportion of the global burden of disease [1]. Globally, the prevalence of dementia among the population of >60 years of age is 5–7%, with 35.6 million people suffering from dementia in 2010, a figure that duplicates every 20 years approximately [2]. Additionally, it is one of the main causes of disability among the elderly and it is the second leading cause of death in high-income countries [3]. Therefore, any progress in the prevention of dementia, which the World Health Organization considers a key factor for countering this pandemic, can benefit millions of people [4].

Some countries, such as the USA, have estimated that delaying the onset of Alzheimer's disease by five years in the population would reduce the prevalence of dementia by 1.2 million people ten years after the intervention [5]. Other calculations show that reducing by 10% each of the relevant risk factors per decade would yield a reduction of 8.3% in the expected rate of Alzheimer's disease [6], similarly to other estimates in Europe [7] and Australia [8].

Several studies have reported diabetes, high blood pressure, obesity, tobacco addiction, depression, low educational level, and physical inactivity as risk factors for dementia and Alzheimer's disease [9,10]. This has promoted research in the field of preventing dementia, such as the collaboration of research groups in the European Dementia Prevention Initiative [11]. A recent study considered that around 35% of cases of dementia can be attributed to nine risk factors: low educational level, hypertension and obesity at middle ages, hearing loss, depression, diabetes, physical inactivity, tobacco addiction, and social isolation [12].

Of note, most studies on modifiable risk factors and dementia are observational, and the few existing clinical trials show disparate results. A randomized clinical trial reported that a

multi-factor intervention (diet, physical exercising, cognitive training, and CVR control) improved or maintained the cognitive function of the elderly with high cognitive risk [13], but other clinical trials have failed to demonstrate a reduction in the prevalence of dementia or cognitive deterioration [14,15].

Current data indicate that CVR factors increase the risk of suffering cognitive deterioration. Diabetes [16,17], tobacco use [18], physical activity [19], high blood pressure [20], obesity [21], hyperlipidemia [22], and auricular fibrillation [23,24] are the most assessed factors. The prevalence of these CVR factors in Spain is high [25,26], even though some studies show low coronary risk in the population [27,28]. Genetic and environmental factors have been suggested to account for these differences [29,30], which has been described as the so-called "Mediterranean paradox" or "French paradox", where the prevalence of coronary disease for Southern European countries is lower than that expected [29]. The prevalence of brain stroke in Spain is also known to be lower than that in other countries [31], and it appears that greater adherence to the Mediterranean diet can decrease the risk of cognitive deterioration and dementia [32]. The relationship between CVR factors and cognitive performance has not been sufficiently investigated in the Spanish population and confirming their role as risk factors could help in the prevention of dementia.

Chronic conditions usually present long, asymptomatic latency periods. This is an advantage for their prevention and treatment since interventions for delaying their onset can drastically reduce the burden for society. Identifying which individuals are at higher risk of suffering cognitive deterioration in subsequent years by using models for predicting CVR can be a useful approach for developing strategies for the prevention of cognitive impairment. Estimating CVR through different risk functions, such as the Framingham's risk score or the Systematic Coronary Risk Evaluation (SCORE), has proven its usefulness for predicting cognitive performance in subsequent years [33–37], finding that greater scores were related poorer performance in neuropsychological tests. However, this has not been sufficiently studied in the Spanish population.

The study NEurological DIsorders in CEntral Spain (NEDICES) [38] commenced in 1993 and consisted of a population, longitudinal study including a total of 5,278 participants ≥65 years of age, with a follow-up from 1994 to 2008, and with two types of objectives: neurologic and general. In 2011, the study NEurological DIsorders in CEntral Spain with biobank (NEDICES2) [39] began, which included younger subjects (≥55 years) and incorporated the services of a biobank (blood, urine, saliva, and hair). This cohort was meant to assess risk factors and biomarkers of age-related neurologic conditions.

NEDICES2-RISK is a new project enrolling patients 55 to 74 years of age from a former study NEDICES2 who underwent medical and neuro-psychological evaluation between 2014 and 2017 and did not present dementia. The project NEDICES2-RISK will evaluate the CVR of the included population and assess the association between CVR at baseline and the change in cognitive performance in the subsequent years.

Additionally, the relationship between these risk factors and indicators of cognitive reserve (CR) [40] is relatively unknown. Mainly, the CR consists of the ability of individuals to optimize execution based on a more efficient use of brain networks. The variability in CR can be due to genetic differences and/or events experienced throughout life such as education, intelligence quotient, occupation, or leisure activities. Previous work by this research team showed that certain factors related to CR (education and occupation) and regular physical activity act as protective factors against developing dementia [41,42].

The aim of the study NEDICES2-RISK is to investigate the association between CVR and the change in cognitive performance after a 5-year follow-up. For this, the REGICOR (Registre Gironí del Cor) [43] and FRESCO (Función de Riesgo Española de acontecimientos

Coronarios y Otros) [25] equations for estimating CVR, which are based on the Framingham's functions and validated in the Spanish population, will be used together with the brief neuro-psychological test developed by the study NEDICES2. Participants that had suffered a cardio-vascular event before the beginning of the study are already classified as having a very high CVR and therefore CVR equations will not be estimated for them. Indicators of CR and life-style will be also evaluated as potential mediating factors that can influence the relationship between CVR and changes in different cognitive functions.

At the moment this manuscript is being produced, screening the general population for cognitive deterioration is not recommended [44], contrary to estimating CVR, which is com-monly performed at primary care consultations. Proving that high CVR implies worse cogni-tive performance in the Spanish population can involve improvements in the field of preventing dementia and allow for defining the optimal target population for intervention strategies. Relying on studies in the Spanish population to back the relationship between CVR factors and future cognitive performance can affect how to tackle such factors and the design of more efficient prevention strategies in Spain.

## Objectives

### Primary aim

To determine the relationship between CVR, estimated in primary prevention using the Fra-mingham-REGICOR and FRESCO risk equations, and the change in cognitive performance after 5 years in the subjects included in NEDICES2-RISK study.

### Secondary aims

a. To describe the profile of patients based on their CVR.

b. To assess the association between each of the studied CVR factors and the change in cogni-tive performance after 5 years.

c. To assess the association between CVR, as estimated in primary prevention using the REGI-COR and FRESCO risk equations, and other non-neurological diseases.

d. To analyze potential mediating factors between CVR, cognitive performance and lifestyle.

e. To assess the effect of indicators of CR (verbal intelligence, education, occupation, and life-style) on cognitive performance, as measured via different neuropsychological tests.

## Methods/Design

### Design

Observational, analytic, prospective cohort study with a 5-year follow-up.

### Ambit

Patient recruitment took place in the primary care setting within the Spanish National Health System in the regions of Ávila, Madrid, Salamanca, and Segovia, Spain. The first evaluation of patients was conducted between 2014 and 2017 and the second examination commenced in 2020 and is expected to end in 2022.

## Population

The studied population are patients 55 to 74 years old, already included in the cohort of the former study NEDICES2 (2014–2017), and not showing dementia in the course of the medical and neurological evaluation. The population in the study NEDICES2 originated from the list of users (social security card holders) of the participating doctors in the included healthcare centers, who were then selected following randomized sampling of patients ≥55 years-old stratified by gender and age (5-year intervals) [39].

## Selection criteria

1. Criteria for inclusion

   - Age from 55 to 74 years

   - Providing informed written consent for their inclusion in the study

   - Having undergone the neurological test in the study NEDICES2

2. Criteria for exclusion

   - Presenting a diagnosis of dementia at baseline

## Sample size

The sample will include all subjects 55 to 74 years of age who were recruited in the cohort NEDICES2 (2014–2017) and underwent basal neuropsychological examination not showing dementia (N = 962).

   With the number of subjects available (N = 962), the power to estimate an effect size of 0.6 when comparing the means of the score of the brief neuropsychological test among the CVR groups, with a sample size ratio of 4.5 and a 95% confidence will be 81%.

## Measurements

1. The main variables and measurements recorded during the study will be:

   a. Socio-demographic: age, gender, educational level, current occupation, marital status, weight at birth, family unit, and family background.

   b. Health habits: sleep hours, alcohol consumption, CAGE questionnaire, caffeine consumption, tobacco consumption, physical activity (measured using the General practice physical activity questionnaire, GPPAQ), Mediterranean diet (measured using the Mediterranean lifestyle index, MEDLIFE index), and social relations.

   c. Health-condition questionnaires: self-reported health and health status (measured using the European Quality of Life Scale 5D, EuroQol-5D).

   d. Anthropometric and physical evaluation: weight, height and blood pressure.

   e. Diseases and treatments: head trauma, ictus, heart disease, atrial fibrillation, hypertension, hypercholesterolemia, diabetes, obesity, memory loss, dementia, Parkinson's disease, Alzheimer's disease, tremors, orofacial dyskinesia, depression, anxiety, other psychiatric disorders, cancer, renal failure, disorders of the digestive system, chronic respiratory condition, rheumatic disease, eye disease, earing deficit, hearing aid, fractured hip, endocrine disorders and chronic pharmacological treatment.

f. Neuropsychological evaluation: Center for Epidemiological Studies Depression Scale (CES-D), 37-item version of the Mini-Mental State Examination (MMSE-37), immediate and differed memory measured via the SEN-FIS illustrations, 11-item version of Pfeffer's Functional Activities Questionnaire (FAQ), Katz index of Independence in Activities of Daily Living (ADL), Trail Making Test, oral fluency test by categories (category: animals), word stress test, and clock-drawing test.

g. CVR estimation: Framingham-REGICOR and FRESCO risk functions in primary prevention.

Table 1 shows the different tools for the collection and measurements of variables.

## Measurement tools

a. Anthropometric measurements: validated and calibrated tools at the primary care consultations employed in usual clinical practice.

b. General questionnaires:

a. CAGE test [45] for detecting risky alcohol use. It comprises four questions (Cut-down, Annoyed, Guilty, Eye-Opener) with a yes/no answer.

**Table 1. Variables and measurements recorded during the study.**

|  | Variables recorded at baseline | Variables recorded at 5-year follow-up |
|---|---|---|
| Socio-demographic variables collected by the interviewer: | X | X |
| • Age | X | X |
| • Gender | X |  |
| • Marital status | X |  |
| • Educational level | X |  |
| • Occupation |  |  |
| Health-related variables recorded by doctors: |  |  |
| • Chronic conditions | X | X |
| • Chronic treatments | X | X |
| • Anthropometric measurements | X | X |
| Lifestyle and health status variables recorded by the interviewer: | X | X |
| • Alcohol consumption | X | X |
| • Caffeine consumption | X | X |
| • Tobacco consumption | X | X |
| • Physical activity | X | X |
| • MEDLIFE index | X | X |
| • EuroQoL 5D |  |  |
| • Quality of life scale |  |  |
| Neuropsychological scales and tests conducted by the interviewer: | X | X |
| • CES-D | X | X |
| • Word stress test | X | X |
| • MMSE-37 | X | X |
| • Pfeffer's FAQ | X | X |
| • Oral fluency test | X | X |
| • SEN-FIS's memory illustrations | X | X |
| • Trail Making Test | X | X |
| • Clock-drawing test |  |  |
| Exposure variables estimated by the research team: |  |  |
| • Framingham-REGICOR | X | X |
| • FRESCO | X | X |

 b. General practice physical activity questionnaire (GPPAQ) [46] for physical activity. Self-administered questionnaire for adults comprising three questions that classifies patients in one of four levels of physical activity.

 c. European Quality of Life (EuroQol-5D) scale [47]. Questionnaire that measures health-related quality of life and can be self-administered. It explores five dimensions (mobility, self-care, daily-life activities, pain/malaise, and anxiety/depression) with three different levels of severity.

 d. Mediterranean lifestyle (MEDLIFE) index [48]. Questionnaire developed for assessing adherence to the Mediterranean lifestyle. It contains 30 items in three blocks: eating Mediterranean food (15 items); Mediterranean dietary habits (7 items); and physical activity, rest, and social interaction habits (8 items). Each item scores 1 or 0 for reaching or complying with an established limit or not, so that the total score ranges from 0 to 30, with higher scores indicating greater adherence to the Mediterranean lifestyle.

c. Protocol of neuropsychological evaluation:

 a. Center for Epidemiological Studies Depression Scale (CES-D) [49]. Self-administered screening tool for detecting depression that records relevant symptoms and its different expressions in the last seven days, classifying each of them in four levels of frequency.

 b. Mini-Mental State Examination, 37-item version (MMSE-37) [50–53]: Spanish version that has been extended, modified, and validated. It evaluates time and space orientation, memory, attention, calculation, language, objects recognition, basic orders, and visual-constructional capacity. This version presents three modifications to Folstein's original MMSE: drawing two intertwined circles is added for assessing visual-constructional capacity, which is easier than making pentagons for the illiterate population; for orders execution, an additional visual task is set consisting of copying an illustration depicting a person with arms raised; for the attention assessment, there is a new task consisting of repeating five numbers (1-3-5-7-9) in inverse order, which is considered to be easier for people of low education than the inverse spelling of a word. For the item assessing the difficulty of repeating three challenging words, the chosen sentence was "en un trigal había tres tigres".

 c. Immediate and deferred memory using illustrations by the Sociedad Española de Neurología (SEN) [54–56]. It explores objects recognition, immediate memory, and deferred memory at five minutes by showing six illustrations depicting common objects.

 d. The 11-item version of Pfeffer's Functional Activities Questionnaire (FAQ) [57,58]. Questionnaire administered to the caregiver that assesses 11 activities involving the use of tools in daily life and competences in the home, as well as occupational and social functioning. It classifies each of the 11 items in three functional levels.

 e. Katz index of Independence in Activities of Daily Living (ADL) [59]. Questionnaire administered to the caregiver to explore basic activities of daily life. It comprises five sections with three possible responses.

 f. Trail Making Test (TMT), series A [60,61]. Test of visual-motor integration that records score and time for execution.

 g. Oral fluency test by categories. It assesses semantic memory, execution capacity, and verbal planning. There are brief versions, such as one noting the number of elements in a

category (animals) [62,63] that the subject can evoke as quickly as possible in one minute.

h. Word stress test [64]. Spanish adaptation of the Adult Reading Test [65]. Test for oral intelligence, where the participant reads 30 non-common words without stress marks and must place the stress on the correct syllable. This test estimates the premorbid intellectual level of the subject.

i. Clock-drawing test. Brief test to explore comprehension, concentration, visual memory, abstraction, and visual-constructional apraxias [66].

d. CVR estimation: risk functions are used for estimating overall coronary risk in primary prevention. These tools stem from studies with cohorts and follow-up periods >10 years and allow for estimating the onset of such diseases over that follow-up. In Spain there are two validated functions:

a. Framingham-REGICOR [43]: Framingham-Wilson's function, calibrated by the REGI-COR group with data from Northeast-Spain population, that is based on the following CVR factors: age, gender, blood pressure, diabetes mellitus, tobacco habit, and cholesterol.

b. FRESCO[25]. A tool developed with data from 12 Spanish cohorts with a simplified model (age, tobacco use, and body mass index) and a complete model (age, tobacco use, diabetes, systolic blood pressure, total cholesterol, HDL cholesterol, interaction of hypertension treatment and systolic blood pressure >120 mmHg, interaction of age and tobacco use, and interaction of age and systolic blood pressure).

## Methods

Patients from the study NEDICES2 without dementia, 55 to 74 years of age, and who were examined between 2014 and 2017 will be contacted by phone 5 years after the first evaluation. Health staff from their healthcare center will inform the patients of the purpose of the study by telephone. Patients who accept to participate and sign informed consent will be evaluated by trained interviewers using the same neuropsychological test they underwent at baseline (conducted between 2014 and 2017) at a five-year follow-up. Healthcare workers will record anthropometric variables, morbidity, CVR factors, and chronic pharmaceutical treatment. The evaluation will take place at each patient's health center or at their home in the cases of immobile subjects. The interviews with the health workers and interviewers are estimated to last 15 minutes and 60 minutes, respectively.

Up to date, 361 patients have been recruited and undergone medical and neuro-psychological evaluation as shown in Fig 1.

## Ethics approval and consent to participate

The protocol was approved (February 12th of 2019) by the Ethics Committee for Research of the Hospital Universitario 12 de Octubre (CEIm 19/021) and was favorably approved by the Central Board for Research of the Primary Healthcare Management of Madrid (February 20th of 2019).

The study will be conducted in accordance with Spanish Law 14/2007 (July 3th of 2007) on biomedical research and the last update of the Helsinki Declaration (Fortaleza, 2013). Patients provided informed consent for their inclusion in the NEDICES2 study, which will be obtained again for the NEDICES2-RISK study.

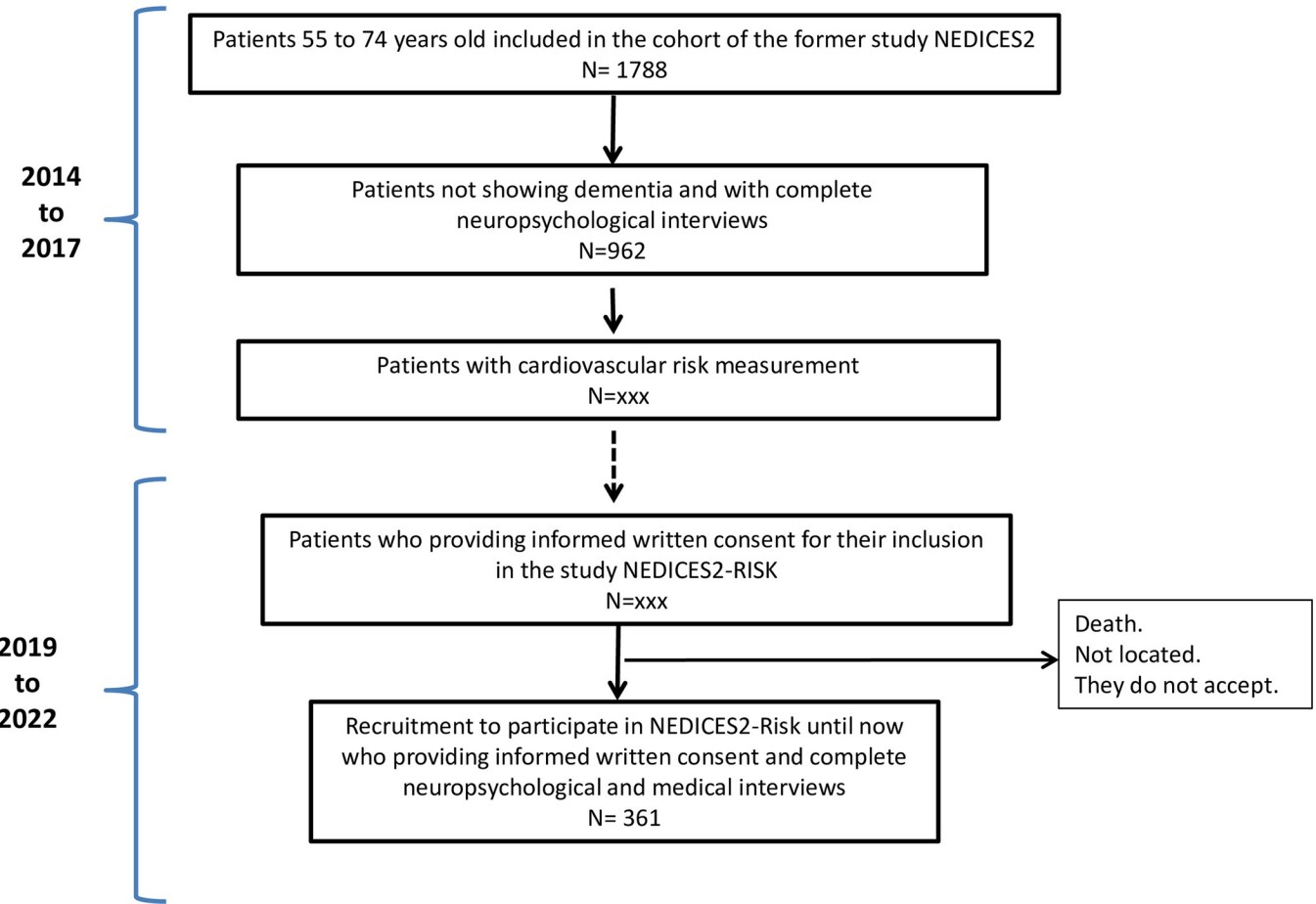

**Fig 1. Flow chart of the study with the number of subjects included.**

All personal and clinical data will be treated in compliance with current law, specifically Spanish Law 3/2018 (December 5th of 2018) on Personal Data Protection and Guarantee of Digital Rights, and Regulation (UE) 2016/679 (April 27th of 2016) by the European Parliament and Council of the European Union on General Data Protection (GDPR).

A local database will be created to record all the obtained data. Upon finalization of this study, this database will be sent to the central database of the NEDICES2 study, and one copy will be kept at the Department of Neurology of Hospital 12 de Octubre. The nominative database will be protected by passwords only available to the researchers.

The estimations of CVR and MMSE-37 will be handed over to the primary healthcare physicians responsible for the included patients in order to adjust their treatment if necessary or further their evaluation of cognitive performance.

## Statistical analysis

The quality of data will be checked at baseline to detect codification errors, missing data, and potential biases related to the representativeness of the sample. Losses to follow-up and abandonments will be recorded together with the characteristics of the patients and reasons for it, and these patients will be compared with those who complete the follow-up.

Characteristics of patients will be described by their frequencies and percentages in the cases of categorical variables, and by their means (standard deviation) or median (interquartile

range) in the case of quantitative variables. An overall cognitive score will be obtained from all cognitive tests: first, the raw scores from each test will be standardized to Z-scores (mean = 0, SD = 1) using the basal mean and SD values in the entire cohort; then, the Z-scores will be averaged to yield the overall cognitive score. The standardized scores will be used to allow for comparisons between items across the test.

An analysis of covariance will be employed to assess the association between basal CVR and cognitive performance at the 5-year follow-up as measured by the different psychometric and functional tests, and linear regression multilevel mixed-effects models will be adjusted to account for intra-subject variability (by taking repeated measures) and clustering of patients (due to recruitment by healthcare center). A multilevel linear regression model will be constructed where the dependent variable will be the score for each of the conducted tests (one model per scale) and the independent variable will be the basal CVR adjusted by the relevant socio-demographic (age, sex and educational achievement) and clinical variables and potential confusion factors.

Additionally, the reliable change will be analyzed for each test to check which clinically significant changes occur in subjects with high CVR. The reliable change will be assessed using the method by Hsu [67], whereas the methods by McSweeny [68] and Duff [69] will be employed to obtain simple and complex rules based on linear regression to calculate the cutoff points of change adjusted by age, gender, and education level (confidence intervals set at 90% and 95%). Data analysis will adhere to established standards of diagnostic accuracy [70] and statistical significance will be set at $p<0.05$.

## Discussion

NEDICES2-RISK is a longitudinal, prospective, observational cohort study whose aim is to estimate the relationship between CVR and cognitive performance at a 5-year follow-up. Among the strengths of this trial is the participating sample, since subjects have been selected from the general population and not based on health characteristics. Additionally, the age of the included subjects (>54 years) is lower than that in most studies evaluating the effect of CVR factors on cognitive performance (average age >75 years). Including younger patients is essential for assessing how CVR factors can affect the onset of the neurodegenerative process at very early stages, which will later develop into cognitive deterioration and dementia.

A potential limitation of the study is that not all the population included in the NEDICES2 study underwent neuropsychological examination due to the lack of economic resources to complete neuropsychological interviews, so they are not eligible for the study NEDICES2--RISK. Although this could result in selection bias, no differences have been found after comparing the basal characteristics of these two groups of patients in terms of age, gender, current smokers, presence of obesity or diabetes, or previous vascular illness (ischemic heart disease and ictus). However, the percentage of subjects with hypercholesterolemia and high blood pressure was higher in the NEDICES2-RISK cohort (Table 2). Despite these differences, the studied sample can be considered as representative of the general population.

The variable of exposure (CVR) will be calculated with the data specifically recorded in the study NEDICES2. However, information from the clinical file of patients will be used in the cases of missing data. Completing data from this secondary source of information can affect the quality of data of variables like blood pressure, weight, height, or lifestyle, since they may be also missing or because of the potentially increased variability resulting from multiple professionals collecting them. However, working with electronic clinical files, a system that has been implemented for over 15 years, guarantees the quality and homogeneity of the data they contain.

**Table 2. Characteristics of the population in the study NEDICES-2.**

| | Without neuropsychological examination (n = 824) | With neuropsychological examination (NEDICES-2-RISK)(n = 962) | p |
|---|---|---|---|
| **Age [Mean (years old) ± SD]** | 66.26 ±5,5 | 65.30 ± 5.5 | 0.051[a] |
| **Gender [n (%)]** | | | 0.730 [b] |
| **Men** | 383 (46.4%) | 455 (47.3%) | |
| **Women** | 441 (53.5%) | 507 (52.7%) | |
| **CVRF and CVD [n (%)]** | | | |
| **Referred obesity** | 112 (17.9%) | 214 (22.2%) | 0.059 [b] |
| **Body mass index >30** | 87 (27.9%) | 297 (30.9%) | 0.204[b] |
| **Hypercholesterolemia** | 248 (39.6%) | 493 (51.2%) | 0.000 [b] |
| **Diabetes** | 92 (14.7%) | 182 (18.9%) | 0.066 [b] |
| **High blood pressure** | 262 (41.9%) | 461 (47.9%) | 0.018 [b] |
| **Current smoker** | 92 (11.2%) | 136 (14,1%) | 0.060[b] |
| **Cardiovascular events** | 41 (6.5%) | 76 (7.9%) | 0.314[b] |

[a] *T*-test

[b] Chi-squared test; CVRF (cardiovascular risk factors); CVD (cardiovascular disease).

CVR factors can change throughout the studied period, due to no longer being exposed to them or the appearance of new ones. To avoid the error these changes can produce, CVR will be estimated at two time points, namely at the first neuropsychological evaluation and five years later at the second one, and the possible variations between them will be examined.

The potential correlation between CVR and changes in the cognitive performance of the included population in the subsequent years will be analyzed. Observing within the Spanish population that subjects with higher CVR present lower cognitive performance over time compared to subjects with a better cardiovascular profile would improve the design of strategies for the prevention of cognitive deterioration and dementia by allowing to define the optimal target population for interventions. Of note, as part of the policies for the prevention of heart disease, CVR is usually estimated at primary care consultations to help in the decision-making process and adjust the level of the intervention based on the observed risk. Hence, employing estimations of CVR in the prevention of cognitive impairment does not imply utilizing a new tool or additional time at the consultation.

Losses to follow-up are an inherent limitation of this type of studies. Variables related to exposure and disease will be examined to find potential correlations between them and subjects lost to follow-up.

In conclusion, this study can provide information on the relationship between CVR, as calculated in usual clinical practice conditions, and the risk of developing cognitive deterioration, which can help in the design of strategies for preventing cognitive impairment by controlling modifiable risk factors.

## Supporting information

**S1 Checklist. STROBE statement—checklist of items that should be included in reports of *cohort studies.***
(DOCX)

**S1 File.**
(DOCX)

**S2 File.**
(DOCX)

## Acknowledgments

We would like to thank Isabel del Cura González as well as the rest of staff in the Research Unit, for their contribution to the design of the project, writing of this manuscript, and their support during the stay for research activity intensification that took place at their premises.

We are grateful to all professionals participating in the NEurological DIsorders in CEntral Spain (NEDICES2 study): participating Primary Care Research Unit of Salamanca (José Luis Alberca- Herrero, Concepción M. Becerro-Muñoz, Lucas Fernández-del Campo, Susana González- Sánchez, Mercedes Meigide-García, Ana Rosa Menor-Odriozola, Sara Mora-Simón, M. Paz Muriel-Díaz, Emiliano Rodriguez-Sánchez, J Alfonso Romero-Furones, Olaya Tamayo-Morales, Jaime Unzueta-Arce); Healthcare centre Arévalo, Ávila (Teresa Sobrino-Arroyo, Pilar Marqués-Macías, Ana Benito-Pérez, Candelas Teresa Martín-García Sancho, Modesta Mulero-San José, Laureano López-Gay, Margarita Jiménez-Nieto, Cristina López-Enríquez, María Antonia Jiménez-Carabias, Elena Donis-Mulero; Ruth Blanco Marqués; Joaquín López-Hebrero); Healthcare centre Cantalejo, Segovia (Juan Francisco Gil-García, César Sanz-Herrero, María Teresa Calvo-Navajo, Ana María de Lucas-Herrero, Alfonso García-Luengo, Laura Rincón-Heras, Eva María Alvárez-de-Castro, Elvira Compes-Ribes, María José Monjas-Heras); Healthcare centre Las Calesas, Madrid (Diego Martín-Acicoya, Nieves Calvo-Arrabal), Healthcare centre Comillas, Madrid (Raquel Calvo-Müller, Pilar Romera-Gutiérrez, María Amparo Pilar Hernando-García), and Healthcare centre Fuentelarreina, Madrid (María Luisa Asensio-Ruiz, María Concepción Díaz-Laso, María Victoria Díaz-Puente, Rafael Ruiz Morote-Aragón, José María Vizcaíno Sánchez-Rodrigo).

## Collaborators

NEDICES2-RISK group collaborators by regions: Salamanca: Luís García-Ortiz[1], Manuel Ángel Gómez Marcos[1], Olaya Tamayo-Morales[1], Susana González Sánchez[1], Sara Mora-Simón[1], Jaime Unzueta-Arce[1] Paz Muriel-Díaz[1], Ana Menor-Odriozola[1], Belén Mateos-Montero[1], Alfonso Romero-Furones[1], José Luis Alberca-Herrero[1], Lucas Fernández del Campo-Carranza[1], Lourdes de la Rosa-Gil[1], Mercedes Meigide-García[1], Elena de Dios Rodríguez[1]; Arévalo, Ávila: Pilar Marqués-Macías[2], Ana Benito-Pérez[2], Candelas Teresa Martín-García Sancho[2], Teodoro Moreno-Sobrino[2], Ignacio Conde-Carrillo[2], Rita Morales-Hernández[2], Margarita Jiménez-Nieto[2], Cristina López-Enríquez[2]; Cantalejo, Segovia: Teresa Calvo-Navajo[3], Eugenio Pablo García-de Santos[3], Martín Oswaldo Riofrío-Pastor[3], Marianny del Carmen Guzmán-Jumelles[3], Noelia de la Esperanza-Esteban[3], Cristina Peláez-Martín[3], Elvira Compes-Ribes[3], Madrid: Health Centre Las Calesas: Diego Martín-Acicoya[4]; Health Centre Comillas: Raquel Calvo-Müller[5], Elia Arranz-Martín[5]; Health Centre Fuentelarreina: María Luisa Asensio-Ruiz[6], María Concepción Díaz-Laso[6], María Victoria Díaz-Puente[6], Rafael Ruiz Morote-Aragón[6], José María Vizcaíno Sánchez-Rodrigo[6].

Lead author: Eugenio Pablo García-de Santos

Email: eugeniopablog@gmail.com

[1] Institute of Biomedical Research of Salamanca (IBSAL). Primary Health Care Research Unit, the Alamedilla Healthcare Center, Gerencia Asistencial Atención Primaria, Servicio de Salud de Castilla y León (SACyL), Salamanca, Spain.

[2] Healthcare Centre Arévalo, Gerencia Asistencial Atención Primaria, Servicio de Salud de Castilla y León (SACyL), Arévalo, Spain.

³ Healthcare Centre Cantalejo, Gerencia Asistencial Atención Primaria, Servicio de Salud de Castilla y León (SACyL), Segovia, Spain.

⁴ Healthcare Centre Las Calesas, Gerencia Asistencial Atención Primaria, Servicio Madrileño de Salud (SERMAS), Madrid, Spain

⁵ Healthcare Centre Comillas, Gerencia Asistencial Atención Primaria, Servicio Madrileño de Salud (SERMAS), Madrid, Spain

⁶ Healthcare Centre Fuentelarreina (SERMAS), Gerencia Asistencial Atención Primaria, Servicio Madrileño de Salud, Madrid, Spain

## Author Contributions

**Conceptualization:** Ester Tapias-Merino, María del Canto De Hoyos-Alonso, Israel Contador-Castillo, Teresa Sanz-Cuesta.

**Data curation:** Ester Tapias-Merino.

**Formal analysis:** Ester Tapias-Merino, María del Canto De Hoyos-Alonso.

**Funding acquisition:** Ester Tapias-Merino.

**Investigation:** Ester Tapias-Merino, Teresa Sanz-Cuesta.

**Methodology:** Ester Tapias-Merino, María del Canto De Hoyos-Alonso, Israel Contador-Castillo, Teresa Sanz-Cuesta.

**Project administration:** Ester Tapias-Merino.

**Resources:** Ester Tapias-Merino.

**Software:** Ester Tapias-Merino, María del Canto De Hoyos-Alonso.

**Supervision:** Ester Tapias-Merino, María del Canto De Hoyos-Alonso.

**Validation:** Ester Tapias-Merino.

**Visualization:** Ester Tapias-Merino, María del Canto De Hoyos-Alonso.

**Writing – original draft:** Ester Tapias-Merino.

**Writing – review & editing:** Ester Tapias-Merino, María del Canto De Hoyos-Alonso, Israel Contador-Castillo, Emiliano Rodríguez-Sánchez, Teresa Sanz-Cuesta, Concepción María Becerro-Muñoz, Jesús Hernández-Gallego, Saturio Vega-Quiroga, Félix Bermejo-Pareja.

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
