## [Decision Letter · Decision Letter 0]

11 Jun 2021

PONE-D-21-11979

Cardiovascular risk in subjects over 55 years of age and cognitive performance after five years. NEDICES2-RISK study.

PLOS ONE

Dear Dr. TAPIAS MERINO,

Thank you for submitting your manuscript to PLOS ONE. After careful consideration, we feel that it has merit but does not fully meet PLOS ONE’s publication criteria as it currently stands. Therefore, we invite you to submit a revised version of the manuscript that addresses the points raised during the review process.

We look forward to receiving your revised manuscript.

Kind regards,

Yoshihiro Fukumoto

Academic Editor

PLOS ONE

Journal Requirements:

3. Please include "Study Protocol" in the title of your manuscript.

4. PLOS ONE does not copy edit accepted manuscripts (https://journals.plos.org/plosone/s/criteria-for-publication#loc-5). To that effect, please ensure that your submission is free of typos and grammatical errors.

5. One of the noted authors is a group: NEDICES2-RISK Group.

In addition to naming the author group, please list the individual authors and affiliations within this group in the acknowledgments section of your manuscript.

Please also indicate clearly a lead author for this group along with a contact email address.

6. Please include a separate caption for each figure in your manuscript.

7. Please ensure that you refer to your Figure in your text as, if accepted, production will need this reference to link the reader to the figure.

8. Please include captions for your Supporting Information files at the end of your manuscript, and update any in-text citations to match accordingly. Please see our Supporting Information guidelines for more information: http://journals.plos.org/plosone/s/supporting-information

9. Your ethics statement should only appear in the Methods section of your manuscript. If your ethics statement is written in any section besides the Methods, please move it to the Methods section and delete it from any other section. Please ensure that your ethics statement is included in your manuscript, as the ethics statement entered into the online submission form will not be published alongside your manuscript.

Reviewers' comments:

Reviewer's Responses to Questions

**Comments to the Author**

1. Does the manuscript provide a valid rationale for the proposed study, with clearly identified and justified research questions?

Reviewer #1: Yes

Reviewer #2: Yes

2. Is the protocol technically sound and planned in a manner that will lead to a meaningful outcome and allow testing the stated hypotheses?

Reviewer #1: Yes

Reviewer #2: Yes

3. Is the methodology feasible and described in sufficient detail to allow the work to be replicable?

Reviewer #1: Yes

Reviewer #2: Yes

4. Have the authors described where all data underlying the findings will be made available when the study is complete?

Reviewer #1: Yes

Reviewer #2: Yes

5. Is the manuscript presented in an intelligible fashion and written in standard English?

Reviewer #1: Yes

Reviewer #2: Yes

6. Review Comments to the Author

You may also provide optional suggestions and comments to authors that they might find helpful in planning their study.

Reviewer #1: Major comments:

This manuscript is study protocol. The aim of this study is to investigate the association between CVR and the change in cognitive performance. It is worth working to prevent for cognitive impairment. However there are some concerned in it.

The population in this study are younger such as 54 to 75 years old than that in most studies. Reviewer has doubt about the follow up periods, because there is a possibility that the adequate change to analyze in cognitive performance can not be revealed after only 5-year follow-up among such as relatively young subjects. Authors should reconsider about the follow up period.

Minor comments:

Page 9, line 5: to correct “ available (N=962) “

Table2. Line 2: to correct “Age［Mean (years old) ± SD ］”, 66.26 ± 5.5 and p 0.051

Table2. Line 3: to correct 0.730. The font size of 0.730 is small.

Table2. Line 5: to correct 53.5%.

Table2. Line 8: to correct 27.9% and 30.9%.

Table2. Line 9: to correct 51.2%.

Table2. Line 10 : to correct 0.066

Table2. Line 11: to correct 47.9%.

Table2. Line 12: to correct 11.2%.

Table2. Line 13: to correct 6.5% , 7.9% and 0.314.

Reviewer #2: In this paper, the authors summarized the study protocol of NEDICES2-RISK, in order to examine CVR in Spanish subjects. This paper addresses very important issue. There is one question about the manuscript as described below;

The sample number was set to 962, which may be relatively small sample size as cohort study. Can the authors find novel effective prevention strategies from this study data in five years?

7. PLOS authors have the option to publish the peer review history of their article (what does this mean?). If published, this will include your full peer review and any attached files.

Reviewer #1: No

Reviewer #2: No

---

## [Author Response · Author response to Decision Letter 0]

21 Jul 2021

Responses to editorial requests:

We have revised and modified the manuscript according to your “manuscript body formatting guidelines”.

We have revised and modified the title, authors, and affiliations according to your “title, author and affiliations formatting guidelines”.

The references that have been modified are cited with numbers: 1, 3, 4, 5, 6, 9, 10, 18, 20, 27, 30, 31, 32, 33, 35, 36, 37, 39, 51, 53, 57, 70. References 3 and 4 have been replaced by updated references, and the URLs have been added to the remaining ones.

3. Please include "Study Protocol" in the title of your manuscript.

“Study protocol” has been included in the title of the manuscript.

4. PLOS ONE does not copy edit accepted manuscripts (https://journals.plos.org/plosone/s/criteria-for-publication#loc-5). To that effect, please ensure that your submission is free of typos and grammatical errors.

We have revised the manuscript.

5. One of the noted authors is a group: NEDICES2-RISK Group. In addition to naming the author group, please list the individual authors and affiliations within this group in the acknowledgments section of your manuscript. Please also indicate clearly a lead author for this group along with a contact email address.

We have added the individual authors and their affiliations. We have indicated who the lead author is with his contact email address.

6. Please include a separate caption for each figure in your manuscript.

We have included a separate caption for the figure in the manuscript.

7. Please ensure that you refer to your Figure in your text as, if accepted, production will need this reference to link the reader to the figure.

We have referred to the figure in the text.

8. Please include captions for your Supporting Information files at the end of your manuscript, and update any in-text citations to match accordingly. Please see our Supporting Information guidelines for more information: http://journals.plos.org/plosone/s/supporting-information

There are no supporting-information files that need to be included in the manuscript.

9. Your ethics statement should only appear in the Methods section of your manuscript. If your ethics statement is written in any section besides the Methods, please move it to the Methods section and delete it from any other section. Please ensure that your ethics statement is included in your manuscript, as the ethics statement entered into the online submission form will not be published alongside your manuscript.

The ethics statement has been moved to the Methods section.

Reviewers' comments

Dear reviewers,

Thank you for reviewing the protocol and considering the subject an important object of research. We really appreciate the relevant comments about the trial and understand your concerns about three aspects particularly remarked: participants age, follow-up period, and sample size. Hereafter, we will try to respond to such issues.

Reviewer #1: 

Major comments:

This manuscript is study protocol. The aim of this study is to investigate the association between CVR and the change in cognitive performance. It is worth working to prevent for cognitive impairment. However there are some concerned in it.

The population in this study are younger such as 54 to 75 years old than that in most studies. Reviewer has doubt about the follow up periods, because there is a possibility that the adequate change to analyze in cognitive performance can not be revealed after only 5-year follow-up among such as relatively young subjects. Authors should reconsider about the follow up period.

Participants age was one of the main points of discussion when this project was designed. The study NEDICES (Neurological Disorders in Central Spain) was conducted between 1998 and 2008 and the population included was >65 years of age. However, the study NEDICES 2 (2014–2017), which is the one our trial is based on, included subjects 10 years younger, which allowed for incorporating participants of that age. Different studies have evidenced that cardiovascular risk factors affect cognitive performance and increase the risk of suffering cognitive deterioration and dementia (1–6). Determining at which point these risk factors commence to affect cognitive performance that will show later in life is more complicated, and more trials are required in subjects in their mid-life (7). Therefore, we always considered an advantage to be able to include middle-aged patients. Certain cognitive domains —such as processing speed, work memory, or executive function— begin their deterioration at early ages (8). Our project aimed to study if the cardiovascular risk is one of the related factors and, since the available evidence highlight the importance of cardiovascular risk at mid-life (9,10), we selected a younger population than in other trials.

The reason to limit the inclusion age to 75 years (patients that would be 79-80 years old after a 5-year follow-up) is that the currently available cardiovascular risk tables do not include the population >79 years, which is the age some participants would be after the follow-up period. 

Whether a follow-up and re-evaluation after 5 years is the most adequate choice for a population of this age is another point for reflection, as you have already noted. In the available literature we found great variability in the follow-up periods, from 1 year up to more than 20 years. Dementia generally requires years of evolution before it is established, in occasions decades, but decreased cognitive performance appears long before. Alterations in specific cognitive domains can be detected in trials with 5-year follow-up periods or even shorter (11–20), after which modifications in the cognitive performance of participants can already be found. Since our trial assessed the cognitive function both globally and for different cognitive domains, we expect to find changes in cognitive performance after 5 years.

On the other hand, this study received public funds by the Instituto de Salud Carlos III that allows for the follow-up of participants at 5 years. Hence, the outcome data will be presented at that point. We intend to obtain more funds to continue the follow-up and re-evaluation of the same population, and therefore extend the follow-up period. 

Minor comments:

Page 9, line 5: to correct “ available (N=962) “

Table2. Line 2: to correct “Age［Mean (years old) ± SD ］”, 66.26 ± 5.5 and p 0.051

Table2. Line 3: to correct 0.730. The font size of 0.730 is small.

Table2. Line 5: to correct 53.5%.

Table2. Line 8: to correct 27.9% and 30.9%.

Table2. Line 9: to correct 51.2%.

Table2. Line 10 : to correct 0.066

Table2. Line 11: to correct 47.9%.

Table2. Line 12: to correct 11.2%.

Table2. Line 13: to correct 6.5% , 7.9% and 0.314.

Thank you for the minor comments. We have corrected all the changes noted.

Reviewer #2:

In this paper, the authors summarized the study protocol of NEDICES2-RISK, in order to examine CVR in Spanish subjects. This paper addresses very important issue. There is one question about the manuscript as described below;

The sample number was set to 962, which may be relatively small sample size as cohort study. Can the authors find novel effective prevention strategies from this study data in five years?

In terms of number of participants, the sample included all subjects aged 55 to 74 years who had been previously recruited in the cohort of the study NEDICES2 (2014–2017) and who did not show dementia at a basal neuro-psychological test (N=962). Variability in sample sizes is very large among this type of trials, and the number of subjects varies from slightly more than 200 up to 20,000 (9). Studies with samples similar to or smaller than ours found changes in cognitive performance after similar follow-up periods, which was evident in domains like oral fluency (13), processing speed (16), or cognitive deterioration (18, 21), and in dementia at more advanced ages (22,23). 

Given the age group and sample size of our trial, we do not expect to find many early-stage dementias, but we expect to observe alterations in some domains of cognitive performance. This would enable to hypothesize about which domains are likely to suffer changes in patients with cardiovascular risk and help assess which scales are more appropriate for the early detection of cognitive deterioration.

 1. Plassman BL, Williams JW, Burke JR, Holsinger T, Benjamin S. Systematic review: factors associated with risk for and possible prevention of cognitive decline in later life. Ann Intern Med [Internet]. 2010 Aug 3;153(3):182–93. Available from: http://www.ncbi.nlm.nih.gov/pubmed/20547887

2. Jefferson AL, Hohman TJ, Liu D, Haj-Hassan S, Gifford KA, Benson EM, et al. Adverse vascular risk is related to cognitive decline in older adults. J Alzheimers Dis [Internet]. 2015;44(4):1361–73. Available from: http://www.pubmedcentral.nih.gov/articlerender.fcgi?artid=4336578&tool=pmcentrez&rendertype=abstract

3. Song R, Xu H, Dintica CS, Pan KY, Qi X, Buchman AS, et al. Associations Between Cardiovascular Risk, Structural Brain Changes, and Cognitive Decline. J Am Coll Cardiol. 2020; 

4. Yu J-T, Xu W, Tan C-C, Andrieu S, Suckling J, Evangelou E, et al. Evidence-based prevention of Alzheimer’s disease: systematic review and meta-analysis of 243 observational prospective studies and 153 randomised controlled trials. J Neurol Neurosurg Psychiatry [Internet]. 2020;91(11):1201–9. Available from: http://www.ncbi.nlm.nih.gov/pubmed/32690803

5. de Bruijn RFAG, Ikram MA. Cardiovascular risk factors and future risk of Alzheimer’s disease. BMC Med [Internet]. 2014 Nov 11;12:130. Available from: http://www.ncbi.nlm.nih.gov/pubmed/25385322

6. Gorelick P, Scuteri a, Black S. contributions to cognitive impairment and dementia a statement for healthcare professionals from the American Heart Association/American Stroke Association. Stroke [Internet]. 2011;42(9):2672–713. Available from: http://stroke.ahajournals.org/content/42/9/2672.short

7. Anstey KJ, Ee N, Eramudugolla R, Jagger C, Peters R. A Systematic Review of Meta-Analyses that Evaluate Risk Factors for Dementia to Evaluate the Quantity, Quality, and Global Representativeness of Evidence. Anstey K, Peters R, editors. J Alzheimer’s Dis [Internet]. 2019 Aug 13;70(s1):S165–86. Available from: http://www.ncbi.nlm.nih.gov/pubmed/31306123

8. Hughes ML, Agrigoroaei S, Jeon M, Bruzzese M, Lachman ME. Change in Cognitive Performance From Midlife Into Old Age: Findings from the Midlife in the United States (MIDUS) Study. J Int Neuropsychol Soc [Internet]. 2018 Sep 18;24(8):805–20. Available from: https://www.cambridge.org/core/product/identifier/S1355617718000425/type/journal_article

9. Harrison SL, Ding J, Tang EYH, Siervo M, Robinson L, Jagger C, et al. Cardiovascular disease risk models and longitudinal changes in cognition: a systematic review. PLoS One [Internet]. 2014;9(12):e114431. Available from: http://www.ncbi.nlm.nih.gov/pubmed/25478916

10. Lafortune L, Martin S, Kelly S, Kuhn I, Remes O, Cowan A, et al. Behavioural Risk Factors in Mid-Life Associated with Successful Ageing, Disability, Dementia and Frailty in Later Life: A Rapid Systematic Review. Vol. 11, PLoS ONE. 2016. 

11. Unverzagt FW, McClure LA, Wadley VG, Jenny NS, Go RC, Cushman M, et al. Vascular risk factors and cognitive impairment in a stroke-free cohort. Neurology [Internet]. 2011 Nov 8;77(19):1729–36. Available from: http://www.ncbi.nlm.nih.gov/pubmed/22067959

12. Dregan A, Stewart R, Gulliford MC. Cardiovascular risk factors and cognitive decline in adults aged 50 and over: a population-based cohort study. Age Ageing [Internet]. 2013 May;42(3):338–45. Available from: http://www.ncbi.nlm.nih.gov/pubmed/23179255

13. Brady CB, Spiro A 3rd, McGlinchey-Berroth R, Milberg W GJ. Stroke risk predicts verbal fluency decline in healthy older men: evidence from the normative aging study. The Journals of gerontology. J Gerontol B Psychol Sci Soc Sci. 2001;56(6):340.6. 

14. Kelley BJ, McClure LA, Letter AJ, Wadley VG, Unverzagt FW, Kissela BM, et al. Report of stroke-like symptoms predicts incident cognitive impairment in a stroke-free cohort. Neurology [Internet]. 2013 Jul 9;81(2):113–8. Available from: http://www.ncbi.nlm.nih.gov/pubmed/23783751

15. Reitz C, Tang M-X, Schupf N, Manly JJ, Mayeux R, Luchsinger JA. A summary risk score for the prediction of Alzheimer disease in elderly persons. Arch Neurol [Internet]. 2010 Jul;67(7):835–41. Available from: http://www.ncbi.nlm.nih.gov/pubmed/20625090

16. Carmasin JS, Mast BT, Allaire JC, Whitfield KE. Vascular risk factors, depression, and cognitive change among African American older adults. Int J Geriatr Psychiatry [Internet]. 2014 Mar;29(3):291–8. Available from: http://www.ncbi.nlm.nih.gov/pubmed/23877973

17. Luchsinger JA, Reitz C, Honig LS, Tang MX, Shea S, Mayeux R. Aggregation of vascular risk factors and risk of incident Alzheimer disease. Neurology [Internet]. 2005 Aug 23;65(4):545–51. Available from: http://www.ncbi.nlm.nih.gov/pubmed/16116114

18. Klages JD, Fisk JD, Rockwood K. APOE genotype, vascular risk factors, memory test performance and the five-year risk of vascular cognitive impairment or Alzheimer’s disease. Dement Geriatr Cogn Disord [Internet]. 2005;20(5):292–7. Available from: http://www.ncbi.nlm.nih.gov/pubmed/16166776

19. Elkins JS, O’Meara ES, Longstreth WT, Carlson MC, Manolio TA, Johnston SC. Stroke risk factors and loss of high cognitive function. Neurology [Internet]. 2004 Sep 14;63(5):793–9. Available from: http://www.neurology.org/cgi/doi/10.1212/01.WNL.0000137014.36689.7F

20. Yaffe K, Bahorik AL, Hoang TD, Forrester S, Jacobs DR, Lewis CE, et al. Cardiovascular risk factors and accelerated cognitive decline in midlife: The CARDIA Study. Neurology [Internet]. 2020;95(7):e839–46. Available from: http://www.ncbi.nlm.nih.gov/pubmed/32669394

21. Tervo S, Kivipelto M, Hänninen T, Vanhanen M, Hallikainen M, Mannermaa A, et al. Incidence and risk factors for mild cognitive impairment: a population-based three-year follow-up study of cognitively healthy elderly subjects. Dement Geriatr Cogn Disord [Internet]. 2004;17(3):196–203. Available from: http://www.ncbi.nlm.nih.gov/pubmed/14739544

22. Qiu C, Xu W, Winblad B, Fratiglioni L. Vascular risk profiles for dementia and Alzheimer’s disease in very old people: a population-based longitudinal study. J Alzheimers Dis [Internet]. 2010;20(1):293–300. Available from: http://www.ncbi.nlm.nih.gov/pubmed/20164587

23. Reitz C, Tang M-X, Schupf N, Manly JJ, Mayeux R, Luchsinger JA. Association of higher levels of high-density lipoprotein cholesterol in elderly individuals and lower risk of late-onset Alzheimer disease. Arch Neurol [Internet]. 2010 Dec;67(12):1491–7. Available from: http://www.ncbi.nlm.nih.gov/pubmed/21149810

The figure file has been uploaded to the Preflight Analysis and Conversion Engine digital diagnostic tool in order to ensure that it meets PLOS requirements.

Thanks again for your revision of the protocol. We hope the performed modifications receive your approval. We look forward to your feedback or any new requirement that you may consider necessary.

Yours truly,

Ester Tapias Merino

---

## [Decision Letter · Decision Letter 1]

1 Sep 2022

Cardiovascular risk in subjects over 55 years of age and cognitive performance after five years. NEDICES2-RISK study.

PONE-D-21-11979R1

Dear Dr. TAPIAS MERINO,

We’re pleased to inform you that your manuscript has been judged scientifically suitable for publication and will be formally accepted for publication once it meets all outstanding technical requirements.

Kind regards,

Yoshihiro Fukumoto

Academic Editor

PLOS ONE

Additional Editor Comments (optional):

Reviewers' comments:

Reviewer's Responses to Questions

**Comments to the Author**

1. Does the manuscript provide a valid rationale for the proposed study, with clearly identified and justified research questions?

Reviewer #1: Yes

Reviewer #2: Yes

2. Is the protocol technically sound and planned in a manner that will lead to a meaningful outcome and allow testing the stated hypotheses?

Reviewer #1: Yes

Reviewer #2: Yes

3. Is the methodology feasible and described in sufficient detail to allow the work to be replicable?

Reviewer #1: Yes

Reviewer #2: Yes

4. Have the authors described where all data underlying the findings will be made available when the study is complete?

Reviewer #1: Yes

Reviewer #2: Yes

5. Is the manuscript presented in an intelligible fashion and written in standard English?

Reviewer #1: Yes

Reviewer #2: Yes

6. Review Comments to the Author

You may also provide optional suggestions and comments to authors that they might find helpful in planning their study.

Reviewer #1: All comments have been addressed. Authors replied adequately to my questions.

This reviewer expect to clarify the relationship between CVR and the cognitive performance after a 5 year follow up.

Reviewer #2: This manuscript presents very interesting data. The second manuscript has been revised well. The reviewer has no more comment for the revised manuscript.

7. PLOS authors have the option to publish the peer review history of their article (what does this mean?). If published, this will include your full peer review and any attached files.

Reviewer #1: No

Reviewer #2: No

---

## [Editor Report · Acceptance letter]

15 Nov 2022

PONE-D-21-11979R1 

Cardiovascular risk in subjects over 55 years of age and cognitive performance after five years. NEDICES2-RISK study. Study protocol 

Dear Dr. Tapias-Merino:

I'm pleased to inform you that your manuscript has been deemed suitable for publication in PLOS ONE. Congratulations! Your manuscript is now with our production department. 

Kind regards, 

on behalf of

Dr. Yoshihiro Fukumoto 

Academic Editor

PLOS ONE